# Bidirectional gated recurrent unit network model can generate future visual field with variable number of input elements

**Joohwang Lee**[1], **Keunheung Park**[2], **Hwayeong Kim**[1], **Sangwoo Moon**[1], **Junglim Kim**[3], **Sangwook Jin**[4], **Seunguk Lee**[5], **Jiwoong Lee**[1]*

**1** Department of Ophthalmology, Pusan National University College of Medicine, Busan, Korea, **2** Busan Medical Center, Busan, Korea, **3** Department of Ophthalmology, Busan Paik Hospital, Inje University College of Medicine, Busan, Korea, **4** Department of Ophthalmology, Dong-A University College of Medicine, Busan, Korea, **5** Department of Ophthalmology, Kosin University College of Medicine, Busan, Korea

* alertlee@naver.com

## Abstract

**Data Availability Statement:** The data in this paper contains potentially identifiable or sensitive patient information, and is subject to requirements from the Institutional Review Board. Data is available

### Purpose

This study aimed to predict future visual field tests using a bidirectional gated recurrent unit (Bi-GRU) and assess its performance based on the number of input visual field tests and the prediction time interval.

### Materials and methods

This study included patients who underwent visual field tests at least four times at five university hospitals between June 2004 and April 2022. All data were accessed in October 2022 for research purposes. In total, 23,517 eyes with 185,858 visual field tests were used as the training dataset, and 1,053 eyes with 9,459 visual field tests were used as the test dataset. The Bi-GRU architecture was designed to take a variable number of visual field tests, ranging from 3 to 80, as input and predict visual field tests at the desired arbitrary time point. It generated the mean deviation (MD), pattern standard deviation (PSD), Visual Field Index (VFI), and total deviation value (TDV) for 54 test points. To analyze the model performance, the mean absolute error between the actual and predicted values was calculated and analyzed for glaucoma severity, number of input visual field tests, and prediction time interval.

### Results

The prediction errors of the Bi-GRU model for MD, PSD, VFI, and TDV ranged from 1.20 to 1.68 dB, 0.95 to 1.16 dB, 3.64 to 4.51%, and 2.13 to 2.60 dB, respectively, depending on the number of input visual field tests. Prediction errors tended to increase as the prediction time interval increased; however, the difference was not statistically significant. As the severity of glaucoma worsened, the prediction errors significantly increased.

upon request to the Pusan National University Hospital Institutional Ethics Committee IRB (hskwon@pnuh.co.kr)

**Funding:** This work was supported by the National Research Foundation of Korea grants (No. RS-2023-00247504 to H.K and J.L) and by the Patient-Centered Clinical Research Coordinating Center, funded by the Ministry of Health & Welfare, Republic of Korea (grant no. HC19C0276 to H.K, S. M, and J.L) and by Convergence Medical Institute of Technology R&D project (CMIT2023-04 to H.K and J.L), Pusan National University Hospital. The funders had no role in study design, data collection and analysis, decision to publish, or preparation of the manuscript.

**Competing interests:** The authors have declared that no competing interests exist.

## Conclusion

In clinical practice, the Bi-GRU model can predict future visual field tests at the desired time points using three or more previous visual field tests.

## Introduction

Glaucoma is one of the leading causes of irreversible blindness worldwide, characterized by distinctive structural changes of the optic nerve head (ONH) and retinal nerve fiber layer (RNFL) associated with visual field changes [1, 2]. Detecting the progression of glaucoma as early as possible is important because the already proven treatment methods such as topical medications, laser therapy, and surgery (such as trabeculectomy and valve implantation) to prevent glaucoma progression, and the consequences of an advanced disease state, profoundly affect the quality of life [3, 4]. Automated perimetry is commonly used as a functional test for detecting the progression of glaucoma. However, short and long-term fluctuations as well as inadequate testing frequency are limitations in visual field analysis for detection of glaucoma progression [3].

Recent developments in artificial intelligence and deep learning algorithms have been utilized in various tasks and fields with outstanding performance [5–7]. Recurrent neural networks (RNN) are a type of neural networks that are naturally suited to processing time-series data and other sequential data [8]. Long short-term memory networks (LSTM) and gate recurrent unit (GRU) are two popular variants of RNN with long-term memory [9]. The bidirectional RNN method has been developed via simultaneous training in forward and backward time directions. Lynn et al. demonstrated the effectiveness of bidirectional networks using LSTM and GRU models in a deep RNN network trained with a bidirectional methodology for biometric electrocardiogram identification [7]; showing that bidirectional LSTM and GRU models are more effective than conventional RNN models, and the bidirectional GRU (Bi-GRU) model achieves a relatively higher performance than bidirectional LSTM(Bi-LSTM) model [7].

In our previous study, the RNN-based Bi-GRU demonstrated significantly lower prediction errors in future visual field (VF) compared to linear regression (LR) and LSTM algorithms [10]. In addition, Bi-GRU was the least affected model in terms of worsening reliability indices and glaucoma severity [10]. However, for the 6th VF prediction, only five consecutive VF tests were used as input. The optimal quantity of input data for the RNN model has not been assessed and remains uncertain. In addition, the mean prediction time interval was 0.94 ± 0.73 year, making it difficult to guarantee the prediction performance of VF testing beyond 1 year [10]. Therefore, in this study, we aim to investigate the performance of Bi-GRU to predict future VF based on the variable number of input VF tests and the prediction time interval.

## Materials and methods

All training and test data were obtained from patients who visited the glaucoma clinics at multi-center tertiary hospitals (Pusan National University Hospital, Kosin University Gospel Hospital, Dong-A University Hospital, Busan Paik Hospital, and Pusan National University Yangsan Hospital) between June 2004 and April 2022. All data were accessed in October 2022 for research purposes. No patients overlapped between the training and test datasets. The medical records and ophthalmic examination results of the subjects were retrospectively reviewed,

and the requirement for patient consent was waived by the institutional review boards owing to the retrospective study design. This study was performed according to the tenets of the Declaration of Helsinki and was approved by the Institutional Review Boards of each hospital (Pusan National University Hospital (Approval No.: 2203-018-113), Kosin University Gospel Hospital (Approval No.: 2018-12-028), Dong-A University Hospital (Approval No.: 22–074), Busan Paik Hospital (Approval No.: 2021-03-014-002), and Pusan National University Yangsan Hospital (Approval No.: 05-2018-172)).

## Visual field examination

Automated perimetry was performed by using a Humphrey Visual Field Analyzer 750i instrument (Carl Zeiss Meditec, Inc., Dublin, CA, USA) with the Swedish interactive threshold algorithm (SITA) 24–2 or 30–2. When VF was measured using the 30–2 test pattern, 54 test points (including 2 points of physiologic blind spot) overlapping with the 24–2 test pattern were used. Glaucomatous VFs were those that met at least one of the following criteria: glaucoma hemifield test outside the normal limits and/or pattern standard deviation (PSD) probability outside of 95% of the normal population. The reliability criteria applied were fixation losses (FL) of less than 33%, false-positive (FP) rates of less than 33%, and false negative (FN) rates of less than 33% [11, 12]. The glaucoma severity group was defined based on the patient's last visual field mean deviation (MD). Eyes were further classified as having early (MD > −6 dB), moderate (−6 dB ≥ MD > −12 dB), or advanced (MD ≤ −12 dB).

## Deep learning architecture

We designed an RNN using Bi-GRU cells, the detailed architecture of which is shown in Fig 1. The first layer, input, received a fixed number (80 exams) of visual field exams. Thus, to make it flexible and allow a variety of input VF examinations, we used a special layer, called the masking layer, following the input layer. The masking layer monitors the input data and recognizes a special data format (empty vector) filled with zero values. Once an empty vector is detected, the masking layer sends a signal to all the subsequent layers; thus, the empty vector is ignored. Therefore, the model can accept a variable number of inputs up to maximum 80 examinations, by ignoring the empty vectors.

On top of the masking layer, a single layer of the Bi-GRU consisting of 128 units was connected. Subsequently, a dense layer of 64 units with a hyperbolic tangent function as activation was applied. To prevent overfitting, a dropout layer with a ratio of 0.1 is followed. Finally, a dense layer is connected to the output. The output layer consisted of 57 units for 54 total deviation values (TDVs), MD, PSD and Visual Field Index (VFI). A hyperbolic tangent function is also used for the output layer.

An adaptive moment estimation optimizer (Adam) (learning rate = 0.001) was used, and the mean squared error was used as its loss function. The batch size was 20, and an early stopping function with patience of 20 was used to stop training automatically. The model was developed using Python 3.10 libraries including TensorFlow 2.12.0, Keras 2.12.0, NumPy 1.23.5, and SciPy 1.10.1.

## Deep learning model input data

The input data were 80 vectors consisting of data vectors and the remaining empty vectors. Each vector is a one-dimensional array spanning a length of 115. It contains the time interval values, three reliability indices (FP, FN, and FL), three global indices (MD, PSD, and VFI), 54 pattern deviation values (PDVs) and 54 TDVs. In the case of 2 test points of physiologic scotoma, "0 (zero)" was entered in the input value. The time interval value was defined as the

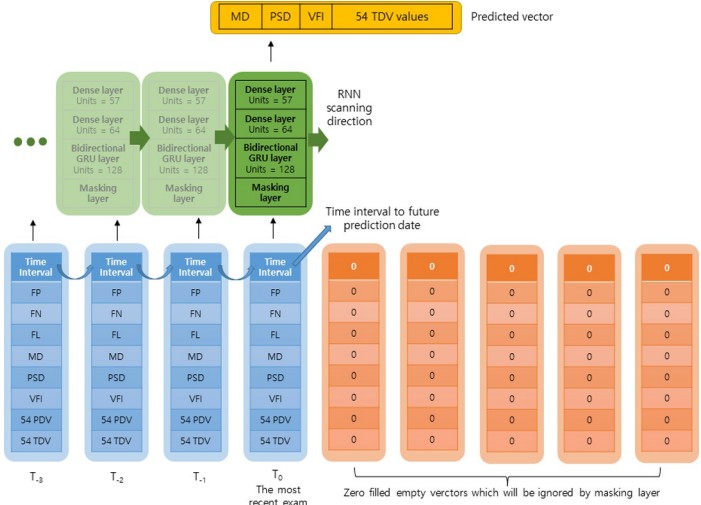

**Fig 1. Architectures of the bidirectional gated recurrent unit (Bi-GRU) model.** Input data comprised 3 categories: relative time displacement in days, reliability data, and visual field (VF) data. To accommodate a variable number of input VF exams, a special layer called a masking layer was used, which monitors the input data and recognizes an empty vector with all values set to "0". This allows the Bi-GRU model to accept a variety of input VF exams up to 80 VF exams by ignoring empty vectors. Reliability data comprised false positive rate (FP), false negative rate (FN), and fixation loss rate (FL). VF data comprised mean deviation (MD), pattern standard deviation (PSD), Visual Field Index (VFI), 54 pattern deviation values (PDVs), and 54 total deviation values (TDVs) of 24–2 standard automated perimetry (including 2 points of physiologic scotoma).

relative temporal distance (day) to the next neighboring visual field examination. In the first data vector, which has no next visual field exam because it is the most recent visual field exam, the time interval value was filled with the temporal distance to the future visual field exam for prediction. For normalizing purposes, the time interval was divided into 1000. All the reliability indices (FP, FN, and FL) were divided by 100. The MD and PSD were divided by 50, and the VFI was divided by 100. The 54 PDVs and 54 TDVs were divided into 50.

## Statistical analyses

To compare the prediction performance, the same accuracy metrics were used as in previous studies, and the root mean square error (RMSE) and mean absolute error (MAE) of the TDV were used as accuracy metrics [12, 13]. The RMSE was calculated for each eye using the following equation:

$$RMSE = \sqrt{\sum_{n=1}^{54} \frac{\left( trueTDV_n - predictedTDV_n \right)^2}{54}}$$

$$n = n^{th} \ test \ point \ of \ visual \ field \ exam$$

$$TDV_n = \ total \ deviation \ value \ of \ n^{th} \ test \ location \ of \ visual \ field \ exam$$

The MAE was also calculated for each eye using the following equation:

$$MAE = \sum_{n=1}^{54} \frac{|\ true\ TDV_n - \ predicted\ TDV_n\ |}{54}$$

$$n = n^{th}\ test\ point\ of\ visual\ field\ exam$$

$$TDV_n = \ total\ deviation\ value\ of\ n^{th}\ test\ location\ of\ visual\ field\ exam$$

For the global indices, including MD, PSD, and VFI, the absolute error between the true and predicted value was calculated for each eye. The Shapiro-Wilk test was performed to check the normality of the data distribution, and nonparametric tests (Mann-Whitney U test and Kruskal-Wallis test) were used to investigate the trends of prediction errors according to various factors such as glaucoma severity, number of input VF tests, and prediction time interval. Statistical significance was set as $P < 0.05$. Statistical analyses were performed using R statistics (version 4.0.3 for Windows).

## Results

The training dataset consists of 23,517 eyes of 12,787 subjects; all subjects underwent at least four VF tests over a mean follow-up duration and age was 4.75 ± 3.58 years and 56.9± 17.0 years, respectively. The VF mean deviation (MD) and mean prediction time interval (the time interval between prediction and the last visual field examination) were −7.81 ± 8.74 dB and 0.84 ± 0.91 years (Table 1). We obtained 195,317 visual fields from 24,570 eyes of 13,348 subjects. A total of 185,858 training dataset records were randomly split into training and validation datasets at a ratio of 9:1. Validation data were used to check the current fitness of the neural network during training to prevent overfitting.

The test dataset consisted of 1,053 eyes of 561 subjects who underwent a minimum of three VF tests. Demographic characteristics are shown in Table 2. The mean follow-up duration and age were 3.52 ± 2.85 years and 51.4 ± 19.0 years, respectively. The mean VF MD and mean prediction time interval were −5.91 ± 7.47 dB and 0.65 ± 0.72 years.

The prediction errors for the global indices and pointwise TDVs according to glaucoma severity are summarized in Table 3. In patients with early glaucoma, the prediction errors for MD, PSD, VFI, TDV MAE, and TDV RMSE were 1.06 ± 1.29 dB, 0.74 ± 0.81 dB, 2.39 ± 3.36%, 1.77 ± 1.22 dB, and 2.36 ± 1.48 dB, respectively. Prediction errors for MD, PSD, VFI, TDV MAE and TDV RMSE increased as glaucoma severity worsened (all $P < 0.001$). Representative examples of VF prediction according to glaucoma severity (early, moderate and advanced stage) based on patient's last VF MD value are shown in S1 Fig.

The prediction errors according to the number of input VF tests and prediction time interval are summarized in Tables 4 and 5, respectively. Depending on the number of input VF tests of the Bi-GRU model, prediction errors ranged from 1.42 ± 1.70 dB to 1.68 ± 2.90 dB for MD, from 0.98 ± 1.06 dB to 1.16 ± 1.46 dB for PSD, from 3.96 ± 6.07% to 4.51 ± 10.01% for VFI, from 2.37 ± 1.55 dB to 2.60 ± 2.64 dB for TDV MAE, and from 3.16 ± 2.06 dB to 3.48 ± 2.97 dB for TDV RMSE. There were no statistically significant differences in prediction errors among the input examination groups (all $P \geq 0.276$) (Table 4). Representative examples of VF prediction according to number of input VF exams and prediction time interval were shown in the S2 and S3 Figs respectively.

Depending on the prediction interval of the Bi-GRU model, prediction errors ranged from 1.33 ± 1.50 dB to 1.97 ± 3.31 dB for MD, from 0.90 ± 0.95 dB to 1.28 ± 1.50 dB for PSD, from

**Table 1. Demographics of patients in the training dataset.**

| | |
|---|---|
| Total number of visual field exams | 185,858 |
| Total number of eyes (patients) | 23,517 (12,787) |
| Age (years), mean ± SD | 56.9 ± 17.0 |
| Follow up period (years), mean ± SD | 4.75 ± 3.58 |
| Visual field mean deviation (dB), mean ± SD | −7.81 ± 8.74 |
| Number of eyes according to glaucoma severity | |
| Early glaucoma | 14,025 |
| Moderate glaucoma | 3,801 |
| Advanced glaucoma | 5,691 |
| Number of eyes according to number of input exams | |
| 3 ∼ 5 exams | 12,161 |
| 6 ∼ 10 exams | 7,347 |
| 11 ∼ 15 exams | 2,599 |
| 16 ∼ 20 exams | 971 |
| 21 ∼ 30 exams | 390 |
| 31 ∼ 40 exams | 34 |
| 41 ∼ exams | 15 |
| Prediction interval (days), mean ± SD | 305.7 ± 331.3 |
| Number of eyes according to prediction interval (months) | |
| < 6 months | 7,946 |
| 6 ≤ ∼ < 12 months | 9,215 |
| 12 ≤ ∼ < 24 months | 4,659 |
| 24 ≤ ∼ < 36 months | 955 |
| 36 ≤ ∼ < 48 months | 362 |
| 48 ≤ months | 380 |

3.41 ± 5.42% to 5.45 ± 12.01% for VFI, from 2.28 ± 1.58 dB to 2.73 ± 3.02 dB for TDV MAE, and from 3.05 ± 2.00 dB to 3.60 ± 2.83 dB for TDV RMSE. There were no statistically significant differences in prediction errors among all prediction interval groups (all $P \geq 0.331$) (Table 5).

The prediction errors of MD, PSD, VFI, TDV MAE and RMSE were binned according to the number of VF input examinations and the prediction time interval are shown in Figs 2 and 3. When the prediction interval exceeded 24 months, the prediction errors for TDV MAE and RSME tended to decrease with a greater number of input VF examinations ($P = 0.065$ and 0.055, respectively, Kruskal-Wallis test). However, there were no statistically significant differences in prediction errors among the input exam groups in all prediction interval groups (all $P \geq 0.055$, Kruskal-Wallis test).

## Discussion

To the best of our knowledge, this is the first study to predict future VFs using an RNN-based Bi-GRU model with a variable number of input VF test and prediction time intervals. Depending on the number of input VF exams of the Bi-GRU model, prediction errors ranged from 1.42 ± 1.70 dB to 1.68 ± 2.90 dB for MD, from 0.98 ± 1.06 dB to 1.16 ± 1.46 dB for PSD, from 3.96 ± 6.07% to 4.51 ± 10.01% for VFI, from 2.37 ± 1.55 dB to 2.60 ± 2.64 dB for TDV MAE, and from 3.16 ± 2.06 dB to 3.48 ± 2.97 dB for TDV RMSE. Although there were no statistically significant differences among input VF exam group and prediction interval groups, the prediction errors for pointwise TDV tended to decrease with greater number of input VF exams

**Table 2. Demographics of patients in the test dataset.**

| | |
|---|---|
| Total number of visual field exams | 9,459 |
| Total number of eyes (patients) | 1,053 (561) |
| Age (years), mean ± SD | 51.4 ± 19.0 |
| Follow up period (years), mean ± SD | 3.52 ± 2.85 |
| Visual field mean deviation (dB), mean ± SD | −5.91 ± 7.47 |
| Number of eyes according to glaucoma severity | |
| Early glaucoma | 748 |
| Moderate glaucoma | 142 |
| Advanced glaucoma | 163 |
| Number of eyes according to number of input exams | |
| 3 ∼ 5 exams | 519 |
| 6 ∼ 10 exams | 245 |
| 11 ∼ 15 exams | 178 |
| 16 ∼ 20 exams | 68 |
| 21 ∼ 30 exams | 38 |
| 31 ∼ 40 exams | 3 |
| 41 ∼ exams | 2 |
| Prediction interval (days), mean ± SD | 237.8 ± 263.3 |
| Number of eyes according to prediction interval (months) | |
| < 6 months | 453 |
| $6 \leq \sim < 12$ months | 431 |
| $12 \leq \sim < 24$ months | 109 |
| $24 \leq$ months | 60 |

when the prediction interval exceeded 24 months (Fig 3). As the severity of glaucoma worsened, the prediction errors significantly increased (Table 3).

Several studies have used artificial intelligence (AI) to detect glaucoma and its progression. Asaoka et al. [14] conducted a study using a deep-learning algorithm to distinguish pre-perimetric glaucoma from normal subjects. The area under the receiver operating characteristic curve (AUROC) was 92.6%, indicating superior diagnostic performance compared to other machine-learning methods. However, the analysis was limited to a small amount of pre-perimetric VF data from 53 eyes and was restricted to classifying VFs rather than predicting future VF. Murata et al. [12, 13] used the variational Bayes linear regression method, a type of machine learning method, to predict the VF progression in patients with glaucoma. Their overall RMSEs based on 5 consecutive input VF data were 4.4 ± 2.5 dB for the Japanese Archive of Multicentral Databases in Glaucoma data and 4.0 ± 2.1 dB for the Diagnostic Innovations in Glaucoma Study data. For our RNN-based Bi-GRU model showed an overall TDV RMSE of 3.24 ± 2.25 dB when the input VF data consisted of 3 to 5 tests, demonstrating superior performance compared to VBLR. Berchuck et al. [15] conducted a study using a deep-learning algorithm known as the generalized variational autoencoder algorithm to predict the progression rate and future VFs of patients with glaucoma. The overall MAE ranged from 1.89 to 2.33 dB, which was comparable to our previous study [10]. Park et al. [11] used an RNN to predict the sixth VF examination using the first five VF tests; they found that the RMSE was 4.31 ± 2.4 dB, indicating that RNN predicted future visual field better than LR. Kim et al. [10] compared future VF examinations using Bi-GRU model with LR and LSTM models. The Bi-GRU model had significantly lower mean prediction errors (RMSE, MAE) compared to LR or LSTM, with values of 3.71 ± 2.42 dB and 2.80 ± 0.36 dB, respectively (all $P < 0.001$).

**Table 3. Prediction errors according to glaucoma severity.**

| Glaucoma severity | No. of eyes | Prediction errors | | | | |
|---|---|---|---|---|---|---|
| | | MD error (dB) | PSD error (dB) | VFI error (%) | TDV MAE (dB) | TDV RMSE (dB) |
| **Early** | 748 | 1.06 ± 1.29 | 0.74 ± 0.81 | 2.39 ± 3.36 | 1.77 ± 1.22 | 2.36 ± 1.48 |
| **Moderate** | 142 | 2.14 ± 1.56 | 1.59 ± 1.59 | 5.38 ± 4.61 | 3.48 ± 1.19 | 4.82 ± 1.74 |
| **Advanced** | 163 | 3.12 ± 3.83 | 1.93 ± 1.63 | 11.08 ± 13.88 | 4.77 ± 3.12 | 6.13 ± 3.29 |
| $P_{all}$ | | <0.001 | <0.001 | <0.001 | <0.001 | <0.001 |
| $P_{early \sim modoerate}$ | | <0.001 | <0.001 | <0.001 | <0.001 | <0.001 |
| $P_{moderate \sim advanced}$ | | 0.174 | 0.0390 | <0.001 | <0.001 | <0.001 |
| $P_{early \sim advanced}$ | | <0.001 | <0.001 | <0.001 | <0.001 | <0.001 |

$P_{all}$: Kruskal-Wallis test among all input exam groups

$P_{early \sim moderate}$: Mann-Whitney U test between the early and moderate glaucoma groups.

$P_{moderate \sim advanced}$: Mann-Whitney U test between the moderate and advanced glaucoma groups

$P_{early \sim advanced}$: Mann-Whitney U test between early ∼ advanced glaucoma group

MD, visual field mean deviation; PSD: Visual field pattern standard deviation; VFI, visual field index; TDV MAE, mean absolute error (MAE) of visual field total deviation values; TDV RMSE, root mean square error (RMSE) of visual field total deviation values.

However, the Bi-GRU algorithm in a previous study could only receive a fixed number of input VF examinations, precluding the generalizability of the algorithm in clinical practice. In this study, we found that regardless of the number of input VF exams, the prediction error of the Bi-GRU model ranged from 2.13 ± 1.81 dB to 2.60 ± 2.64 dB and from 2.80 ± 2.28 dB to 3.48 ± 2.97 dB for TDV MAE and RMSE, respectively. There were no statistically significant differences in the prediction errors for pointwise TDV or global indices among the input VF exam groups, which demonstrated the general application of this Bi-GRU model in real clinical practice. In a previous study using the Bi-GRU algorithm by Kim et al. [10], when the eyes were binned according to prediction error (RMSE), the prediction errors of 2 dB or less accounted for 530 eyes (41.67%), and between 2–3 dB accounted for 175 eyes (13.76%). However, it is important to note that while the study used a fixed set of 5 VF tests as input data to predict the 6[th] VF test, this study used at least three VF tests, and as mentioned earlier, used the masking layer of the Bi-GRU to accommodate a variable number of VF tests as input data, highlighting a significant difference and its implications.

In addition, we did not evaluate the prediction error according to different prediction time interval in the previous study [10]. In this study, prediction errors ranged from 2.28 ± 1.58 dB to 2.73 ± 3.02 dB for TDV MAE, and from 3.05 ± 2.00 dB to 3.60 ± 2.83 dB for TDV RMSE

**Table 4. Prediction error according to number of input visual field exams.**

| No. of input exams | No. of eyes | Prediction errors | | | | |
|---|---|---|---|---|---|---|
| | | MD error (dB) | PSD error (dB) | VFI error (%) | TDV MAE (dB) | TDV RMSE (dB) |
| **3 ∼ 5** | 519 | 1.50 ± 1.87 | 0.98 ± 1.06 | 3.96 ± 6.07 | 2.46 ± 1.87 | 3.24 ± 2.25 |
| **6 ∼ 10** | 245 | 1.68 ± 2.90 | 1.07 ± 1.31 | 4.51 ± 10.01 | 2.60 ± 2.64 | 3.48 ± 2.97 |
| **11 ∼ 15** | 178 | 1.44 ± 1.53 | 1.16 ± 1.46 | 4.16 ± 5.90 | 2.37 ± 1.55 | 3.16 ± 2.06 |
| **>15** | 111 | 1.42 ± 1.70 | 1.01 ± 1.10 | 4.14 ± 5.49 | 2.38 ± 1.71 | 3.19 ± 2.25 |
| *P* value | | 0.694 | 0.276 | 0.682 | 0.653 | 0.807 |

*P* value: one-way ANOVA test among all input exam groups

MD: Visual field mean deviation; PSD: Visual field pattern standard deviation; VFI: visual field index; TDV: mean absolute error (MAE) of visual field total deviation values (TDV); TDV RMSE: root mean square error (RMSE) of visual field total deviation values (TDV).

**Table 5. Prediction error according to prediction interval.**

| Prediction interval (months) | No. of eyes | Prediction errors | | | | |
|---|---|---|---|---|---|---|
| | | MD error (dB) | PSD error (dB) | VFI error (%) | TDV MAE (dB) | TDV RMSE (dB) |
| <6 | 453 | 1.57 ± 2.14 | 1.13 ± 1.30 | 4.41 ± 6.88 | 2.55 ± 2.03 | 3.39 ± 2.43 |
| 6≤ ~ <12 | 431 | 1.33 ± 1.50 | 0.90 ±0.95 | 3.41 ± 5.42 | 2.28 ± 1.58 | 3.05 ± 2.00 |
| 12≤ ~ <24 | 109 | 1.97 ± 3.31 | 1.08 ± 1.40 | 5.45 ± 12.01 | 2.73 ± 3.02 | 3.51 ± 3.31 |
| 24≤ | 60 | 1.76 ± 2.39 | 1.28 ± 1.50 | 4.98 ± 7.02 | 2.69 ± 2.30 | 3.60 ± 2.83 |
| P value | | 0.331 | 0.856 | 0.527 | 0.768 | 0.860 |

P value: one-way ANOVA test among all prediction interval groups

MD: Visual field mean deviation; PSD: Visual field pattern standard deviation; VFI: visual field index; TDV: mean absolute error (MAE) of visual field total deviation values (TDV); TDV RMSE: root mean square error (RMSE) of visual field total deviation values (TDV).

depending on the prediction interval of the Bi-GRU model. We found no statistically significant differences in prediction errors among all prediction interval groups (P = 0.768, P = 0.860, respectively) (Table 5).

Patients with glaucoma were divided into early, moderate, and advanced stages of the disease. We found that the prediction error for the global indices and pointwise TDV increased as glaucoma severity worsened. This finding is consistent with the results of our previous study in which prediction error of Bi-GRU model was negatively correlated with VF MD [10]. The possible explanation for this finding is that VF variability increases with the severity of glaucoma damage [16, 17].

Our study had several limitations. First, an increase in the number of input VF tests did not improve the prediction performance. The reason for this finding may be due to the relatively large proportion of patients with early and stable glaucoma patients (early glaucoma patients with final VF MD > −6 dB, accounting for 14,025 eyes (59.6%) and 1,053 eyes (71.0%) in the training and test dataset, respectively). However, in clinical practice, the vast majority of treated eyes are stable over long period, whereas 3–17% of eyes may be subject to worsening [18]. Eslami et al. also reported that class imbalance in VF test data can cause a decreased in future VF prediction ability in deep-learning models [19]. In this regard, it is necessary to investigate future longitudinal studies to understand how the prediction performance changes

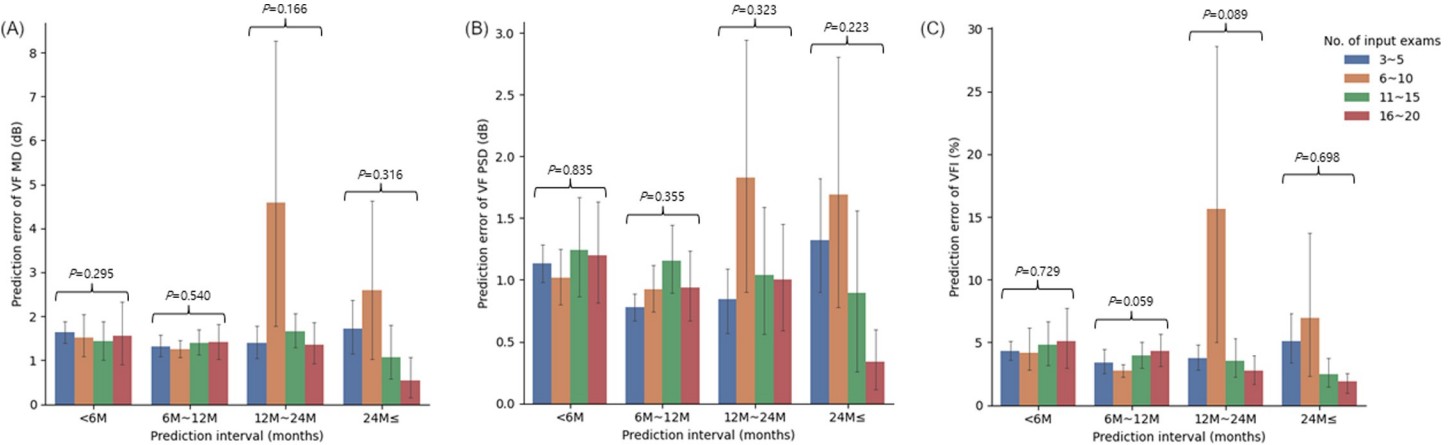

**Fig 2.** Prediction errors of mean deviation (MD) (A), pattern standard deviation (PSD) (B), and Visual Field Index (VFI) (C) were binned according to number of visual field input exams and prediction time interval.

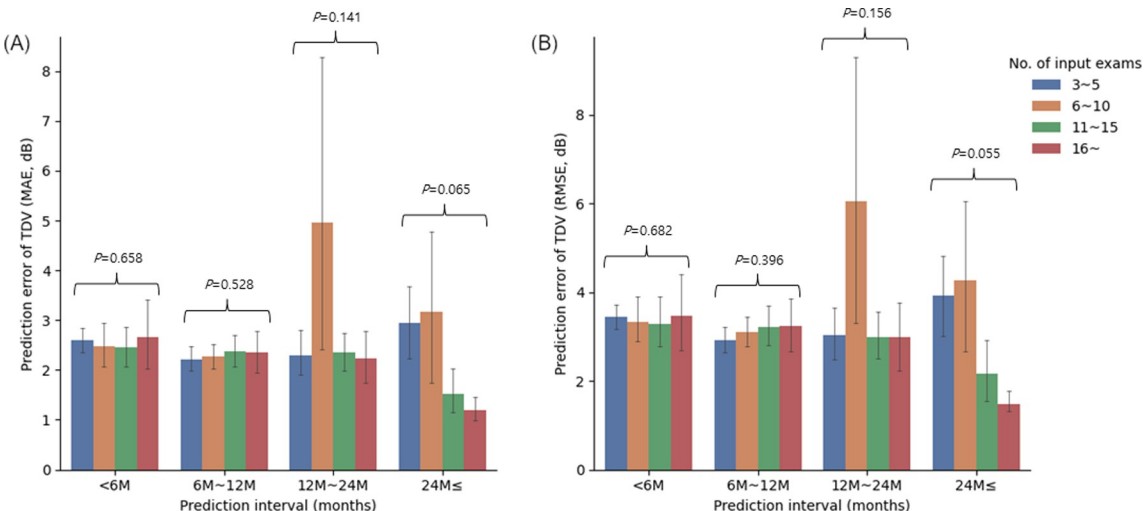

**Fig 3.** Prediction errors of total deviation values (TDV) mean absolute error (MAE) (A) and root mean squared error (RMSE) (B) binned according to the number of visual field input exams and prediction time interval.

in patients with glaucomatous VF progression. Second, we included all available 24–2 VF tests in the training dataset acquired from multiple tertiary hospitals that were not labeled by diagnosis. Consequently, the VF data included not only normal VF data but also data from individuals with non-glaucomatous optic neuropathy, macular degeneration, and retinal vascular disorders. However, this approach may enhance the generalizability of the Bi-GRU model in clinical practice. Third, our study did not include clinical data such as intraocular pressure, central corneal thickness, and structural data. In a previous study by Dixit et al. [20], a convolutional LSTM neural network was employed to detect glaucoma progression using longitudinal VF and including clinical information significantly improved the performance of the algorithm for detecting glaucoma progression. Future research may require the incorporation of clinical characteristics into the input data to enhance the performance of Bi-GRU architectures. Lastly, in this study, the RNN algorithm was trained on a single ethnic group consisting of the sole Asian cohort, which led to data bias (racial bias). To reduce this issue and ensure data generalization, it is crucial to build a diverse and representative training datasets of all races and ethnicities to test and report its performance.

## Conclusion

In this study, we confirmed that the Bi-GRU model can predict future VFs at different time points using more than three input VF tests. This approach has the potential to enable early detection of glaucoma progression in clinical practice, thereby contributing to the prevention of blindness in glaucoma patients.

## Supporting information

**S1 Fig. Representative examples of visual field (VF) prediction according to glaucoma severity (early, moderate and advanced) based on patient's last VF mean deviation (MD) value.** Input VFs are shown in chronological order from left to right, followed by actual VF and the predicted VF of the bidirectional gated recurrent unit model at different future time points. VF total deviation values for each of the 52 test locations are shown in gray scale. (TIF)

**S2 Fig. Representative examples of visual field (VF) prediction according to number of input VF exams.** Input VFs are shown in chronological order from left to right, followed by actual VF and the predicted VF of the bidirectional gated recurrent unit model at different future time points. VF total deviation values for each of the 52 test locations are shown in gray scale.
(TIF)

**S3 Fig. Representative examples of visual field (VF) prediction according to prediction interval.** Input VFs are shown in chronological order from left to right, followed by actual VF and the predicted VF of the bidirectional gated recurrent unit model at different future time points. VF total deviation values for each of the 52 test locations are shown in gray scale.
(TIF)

## Acknowledgments

We would like to thank Editage (www.editage.co.kr) for English language editing.

## Author Contributions

**Investigation:** Hwayeong Kim.

**Software:** Keunheung Park.

**Supervision:** Sangwoo Moon, Junglim Kim, Sangwook Jin, Seunguk Lee, Jiwoong Lee.

**Writing – original draft:** Joohwang Lee.

**Writing – review & editing:** Jiwoong Lee.

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
