## [Decision Letter · Decision Letter 0]

24 Mar 2024

PONE-D-23-36114Bidirectional gated recurrent unit network model can generate future visual field with variable number of input elementsPLOS ONE

Dear Dr. Lee,

Thank you for submitting your manuscript to PLOS ONE. After careful consideration, we feel that it has merit but does not fully meet PLOS ONE’s publication criteria as it currently stands. Therefore, we invite you to submit a revised version of the manuscript that addresses the points raised during the review process.

The authors have drafted very nice manuscript with robust quantitative analysis to support the role of bidirectional gated recurrent unit network model to generate future VF with variable number of input elements. Please scroll down to the bottom of the email for the specific additional feedback.

We look forward to receiving your revised manuscript.

Kind regards,

Natasha Gautam, MBBS, MS

Academic Editor

PLOS ONE

Journal Requirements:

 “This work was supported by the National Research Foundation of Korea grants (No. RS-2023-00247504 and by Convergence Medical Institute of Technology R&D project (CMIT2023-00), Pusan National University Hospital.” 

Additional Editor Comments (if provided):

This is a well written manuscript. I would advise to move the results and details of training dataset including table 1 from methodology to results section, before describing the results about test dataset. Similarly the authors described the criteria to determine glaucoma severity (early, moderate and severe) in results section, which should be written in methodology instead of results.

Reviewers' comments:

Reviewer's Responses to Questions

**Comments to the Author**

1. Is the manuscript technically sound, and do the data support the conclusions?

Reviewer #1: Yes

2. Has the statistical analysis been performed appropriately and rigorously? 

Reviewer #1: Yes

3. Have the authors made all data underlying the findings in their manuscript fully available?

Reviewer #1: Yes

4. Is the manuscript presented in an intelligible fashion and written in standard English?

Reviewer #1: Yes

5. Review Comments to the Author

Reviewer #1: In this study, the authors proposed a deep learning method based on bidirectional gated recurrent units (Bi-GRU) to predict 24-2 Humphrey visual field (HVF) tests. I am impressed by the amount of data the authors used in their training set (23,517 eyes out of 185,858 VF tests from five hospitals from 06/2004 - 04/2022). I also appreciate that the authors clearly stated that the training and test data sets were separated by subject to keep the data sets independent. It is an excellent design that the proposed method can receive a varying number of VFs (up to a maximum of 80 examinations) and is able to predict the mean deviation (MD), pattern standard deviation (PSD), visual field index (VFI), 54 points of pattern deviation value (PDVs), and total deviation value (TDVs).

Overall, this is a well-written manuscript describing a study with an important clinical objective and a straightforward methodology. Here are some suggestions that could further enhance its strengths and clarity.

1. Recurrent neural networks (RNNs) are well-established deep learning (DL) techniques for handling sequential data. However, recent advances have led to the development of newer methods (e.g., transformers) that may provide surprisingly good results. I believe that adding comparisons with a standard (basic) and state-of-the-art method to the test datasets could further validate the results and increase the manuscript's contribution to the field.

2. With such a large number of VF tests, I wonder how the authors digitally access the values in the VF reports. Meanwhile, what are the quality control steps to ensure that the digitally obtained values are the same as those in the HVF reports?

3. The manuscript provides a solid quantitative analysis. Including qualitative results for a few best and worst cases in each prediction category (i.e., glaucoma severity, number of VF tests, and prediction intervals) would substantially provide valuable insights and enrich the narrative and depth of the study.

4. The manuscript demonstrates a thorough analysis using an Asian dataset, which provides valuable insights specific to this population. To enhance the clarity and applicability of the manuscript, I believe it would be beneficial to acknowledge that the focus on this particular dataset may influence the generalizability of the findings to other racial groups. Including a brief discussion of this aspect could further strengthen the study by highlighting areas for future research and the potential application of a more diverse dataset.

6. PLOS authors have the option to publish the peer review history of their article (what does this mean?). If published, this will include your full peer review and any attached files.

Reviewer #1: No

---

## [Author Response · Author response to Decision Letter 0]

31 May 2024

Reviewer’s comment

1. Recurrent neural networks (RNNs) are well-established deep learning (DL) techniques for handling sequential data. However, recent advances have led to the development of newer methods (e.g., transformers) that may provide surprisingly good results. I believe that adding comparisons with a standard (basic) and state-of-the-art method to the test datasets could further validate the results and increase the manuscript's contribution to the field.

Response: Thank you for the comment. In fact, we have already tested “transformer network” as well as BiGRU model. As you mentioned, “transformer network” is a SOTA algorithm and we were very excited to use it. We expected its overwhelming performance just like chatBot does but on the contrary to our expectation, the performance was not so good. Now we are re-trying transformer network to solve the issue. We are also considering to use GTN (Gated Transformer Network) because visual field data contains channel like data (i.e. TDV, PDV, THV). Other than transformer network, we are trying another new architecture, CRU (Correlation Recurrent Units: A Novel Neural Architecture for Improving the Predictive Performance of Time-Series Data, https://ieeexplore.ieee.org/abstract/document/10264112). CRU internally utilizes correlation between input data similar to attention method in transformer. We expect CRU is lighter than transformer but it will show better performance than GRU. Taken together all above methods, we are planning to write a next paper comparing various SOTA architectures including transformer. At the moment, research is still ongoing and we did not yet conclude anything. We are very much thankful of your valuable suggestion but we would be glad if you can wait us to complete the research.

2. With such a large number of VF tests, I wonder how the authors digitally access the values in the VF reports. Meanwhile, what are the quality control steps to ensure that the digitally obtained values are the same as those in the HVF reports?

Response: Thank you for the comment. To obtain such a large amount of visual field data digitally, Zeiss provides a “study mode” in their HVF machine. To activate this mode, you need license key from Zeiss. When study mode activated, you can export all exam data as “XML” files (1 exam per 1 XML file). We developed a custom software to transform XML files to a single excel file which contains all patients’ exam data. Because the manufacturer, Zeiss, guarantees XML file data exported from HVF machine is exactly same as HVF reports, we do not need quality control step. We also confirmed that our custom software converts XML file to excel file without error.

3. The manuscript provides a solid quantitative analysis. Including qualitative results for a few best and worst cases in each prediction category (i.e., glaucoma severity, number of VF tests, and prediction intervals) would substantially provide valuable insights and enrich the narrative and depth of the study.

Response: Thank you for the comment. We have added qualitative results for a representative example for different glaucoma severity, number of input visual field exams, and prediction interval as supplementary materials (S1_fig, S2_fig, S3_fig.pdf including captions and legends) shown as below. and updated the in-text citations.

Early stage

Moderate stage

Advanced stage

S1 Figure. Representative examples of visual field (VF) prediction according to glaucoma severity (early, moderate and advanced) based on patient’s last VF mean deviation (MD) value. Input VFs are shown in chronological order from left to right, followed by actual VF and the predicted VF of the bidirectional gated recurrent unit model at different future time points. VF total deviation values for each of the 52 test locations are shown in gray scale.

Number of input visual field: 3~5

Number of input visual field: 6~10

Number of input visual field: 11~15

Number of input visual field: >15

S2 Figure. Representative examples of visual field (VF) prediction according to number of input VF exams. Input VFs are shown in chronological order from left to right, followed by actual VF and the predicted VF of the bidirectional gated recurrent unit model at different future time points. VF total deviation values for each of the 52 test locations are shown in gray scale.

Prediction interval < 6 months

6 months ≤ Prediction interval < 12 months 

12 months ≤ Prediction interval < 24 months

24 months ≤ Prediction interval 

S3 Figure. Representative examples of visual field (VF) prediction according to prediction interval. Input VFs are shown in chronological order from left to right, followed by actual VF and the predicted VF of the bidirectional gated recurrent unit model at different future time points. VF total deviation values for each of the 52 test locations are shown in gray scale.

4. The manuscript demonstrates a thorough analysis using an Asian dataset, which provides valuable insights specific to this population. To enhance the clarity and applicability of the manuscript, I believe it would be beneficial to acknowledge that the focus on this particular dataset may influence the generalizability of the findings to other racial groups. Including a brief discussion of this aspect could further strengthen the study by highlighting areas for future research and the potential application of a more diverse dataset.

Response: Thank you for the comment. We have added data bias and generalization issues due to the single ethnic group in the Asian dataset to the limitation section. We have added following sentences as shown below. 

“Lastly, in this study, the RNN algorithm was trained on a single ethnic group consisting of Asian, which can lead to data bias (racial bias). To reduce this issue and ensure data generalization, it is crucial to build a diverse and representative training datasets of all races and ethnicities to test and report its performance.”

Sincerely,

Jiwoong Lee, MD, PhD

Department of Ophthalmology, Pusan National University Hospital, 179 Gudeok-ro, Seo-gu, Busan 49241, South Korea

Tel: +82-51-240-7326

Fax: +82-51-242-7341

E-mail: alertlee@naver.com

---

## [Editor Report · Decision Letter 1]

8 Jul 2024

Bidirectional gated recurrent unit network model can generate future visual field with variable number of input elements

PONE-D-23-36114R1

Dear Dr. Lee,

We’re pleased to inform you that your manuscript has been judged scientifically suitable for publication and will be formally accepted for publication once it meets all outstanding technical requirements.

Kind regards,

Natasha Gautam, MBBS, MS

Academic Editor

PLOS ONE
---

## [Editor Report · Acceptance letter]

16 Jul 2024

PONE-D-23-36114R1 

PLOS ONE

Dear Dr. Lee, 

I'm pleased to inform you that your manuscript has been deemed suitable for publication in PLOS ONE. Congratulations! Your manuscript is now being handed over to our production team.

Kind regards, 

on behalf of

Dr. Natasha Gautam 

Academic Editor

PLOS ONE